# Shared latent genetic liability across fibromyalgia and psychiatric traits: Novel insights from genomic structural equation modeling

Liling Lin[1☉*], Yankai Li[2☉], Fengtao Ji[3☉], Jianwei Lin[4], Mengyi Zhu[5], Diefei Liang[6], Minghui Cao[7‡*], Ganglan Fu[8‡*], Yanni Fu[9‡*]

**1** Department of Anesthesiology, Sun Yat-sen Memorial Hospital, Sun Yat-sen University, Guangzhou, China, **2** Zhongshan School of Medicine, Sun Yat-sen University, Guangzhou, China, **3** Department of Anesthesiology, Sun Yat-sen Memorial Hospital, Sun Yat-sen University, Guangzhou, China, **4** Big Data Laboratory, Joint Shantou International Eye Center of Shantou University and The Chinese University of Hong Kong, Shantou, Guangdong, China, **5** Department of Urology Surgery, Sun Yat-sen Memorial Hospital, Sun Yat-sen University, Guangzhou, China, **6** Department of Endocrinology, Sun Yat-sen Memorial Hospital, Sun Yat-sen University, Guangzhou, China, **7** Department of Anesthesiology, Sun Yat-sen Memorial Hospital, Sun Yat-sen University, Guangzhou, China, **8** Department of Anesthesiology, Sun Yat-sen Memorial Hospital, Sun Yat-sen University, Guangzhou, China, **9** Department of Anesthesiology, Sun Yat-sen Memorial Hospital, Sun Yat-sen University, Guangzhou, China

☉ These authors contributed equally to this work and share first authorship
‡ These authors share last authorship
* linll3@mail.sysu.edu.cn (LL); caomh@mail.sysu.edu.cn (MC); fugangl@mail.sysu.edu.cn (GF); fuyanni@mail.sysu.edu.cn (YF)

## Abstract

### Background

Fibromyalgia, insomnia, depression, and anxiety share common clinical comorbidities, but their underlying genetic architecture and mechanism remain unclear.

### Methods

We conducted phenotype-specific Genome-wide association study (GWAS) meta-analyses for fibromyalgia, insomnia, depression, and anxiety, respectively. Genomic structural equation modeling was employed to identify a shared genetic factor (mvFibroPsych). Lead SNPs and associated genes were annotated using Functional Mapping and Annotation (FUMA), followed by gene-set and tissue enrichment analyses. The Latent Causal Variable (LCV) method was utilized to identify modifiable risk factors and phenotypes influenced by mvFibroPsych. Additionally, brain-wide and proteome-wide Mendelian randomization (MR) analyses were applied to explore brain regions and biomarkers associated with mvFibroPsych. Multi-layer molecular quantitative trait locus (QTL) analyses were conducted for mechanistic insights into mvFibroPsych.

**Data availability statement:** The data used in this study were publicly available and can be accessed via the links presented in Table S1 and the references in the manuscript. The datasets supporting the conclusions of this article were included within the article and its additional files. The common factor GWAS of mvFibroPsych generated in this study have been deposited in Figshare andare available at: https://figshare.com/articles/dataset/mvFibroPsych/30464615?file=59113604.

**Funding:** This work was supported by the Medical Science and Technology Research Fund of Guangdong Province, China (Grant number A2024010 to L.L.), National Natural Science Foundation of China (Grant number 82301389 to G.F.), National Natural Science Foundation of China (No. 82300912 to D.L.), and Natural Science Foundation of Guangdong Province (No. 2022A1515111053 to D.L.). The funders had no role in study design, data collection and analysis, decision to publish, or preparation of the manuscript.

**Competing interests:** The authors have declared that no competing interests exist.

## Results

Strong genetic correlations were observed among the four phenotypes ($r_g = 0.55$–$0.84$), with excellent model fit for the common factor [comparative fit index (CFI) = 0.999, standardized root mean square residual (SRMR) = 0.015]. The mvFibroPsych GWAS identified 49 lead SNPs across 43 loci, including 32 novel loci. Gene prioritization revealed 342 protein-coding genes, and pathway analysis indicated enrichment in synaptic function pathway. LCV identified 133 phenotypes causally linked to mvFibroPsych. Brain-wide MR found fractional anisotropy in the splenium of the corpus callosum to be inversely associated with mvFibroPsych. Proteome-wide MR identified five proteins significantly associated with mvFibroPsych, while multi-layer brain QTL analysis prioritized CD40 as a potential target.

## Conclusions

This study provides strong evidence for a shared genetic factor underlying fibromyalgia, insomnia, depression, and anxiety, linked to synaptic function, brain structure integrity, and neuroinflammatory pathways.

### Author summary

Fibromyalgia, insomnia, depression, and anxiety are common health conditions that often occur together, but the genetic factors that connect them remain poorly understood. In this study, we combined data from large-scale genetic studies of these four conditions to identify shared genetic influences. Our analysis revealed a common genetic factor (mvFibroPsych) that underlies these conditions. We identified key genes and pathways, including synaptic function, which may play a role in their shared biological mechanisms. Additionally, we discovered brain region and proteins associated with this latent genetic factor, including markers of brain structure and inflammation. These findings highlight potential areas for developing new treatments targeting these interconnected conditions. By providing insights into how genetic and biological factors contribute to this underlying shared genetic component, our research paves the way for improving diagnosis and treatment for millions of people affected by these overlapping conditions.

## Introduction

Fibromyalgia is a complex, chronic pain disorder characterized by widespread pain, fatigue, and heightened sensitivity to physical stimuli, affecting approximately 2–3% of the global population [1]. Despite extensive research, its etiology remains largely unknown, with evidence suggesting a multifactorial origin that includes genetic, environmental, and psychological influences [2,3]. Notably, fibromyalgia frequently co-occurs with psychiatric disorders, particularly insomnia, depression, and anxiety

[4,5]. Although these psychiatric conditions are well-documented to exacerbate the severity and prognosis of fibromyalgia, their potential shared pathophysiology with fibromyalgia has not been thoroughly explored, despite some emerging evidence [6].

The frequent comorbidity of insomnia, depression, and anxiety with fibromyalgia suggests a potential genetic and neurobiological overlap that warrants deeper investigation. Identifying a shared genetic factor among these disorders could significantly advance our understanding of their underlying biological mechanisms. However, uncovering such common factors poses considerable challenges, as traditional approaches to exploring genetic overlap often require extensive data collection across different cohorts, complex study designs, and substantial financial resources, limiting their feasibility [7,8]. Moreover, the heterogeneous nature of these conditions complicates the precise identification of shared genetic influences, underscoring the need for more accessible and scalable methodologies.

While the psychiatric comorbidities of fibromyalgia are well-documented, its genetic basis remains relatively underexplored. Early candidate gene studies yielded inconsistent findings, and only a few genome-wide association studies have been conducted to date. For example, studies by Docampo et al. and Peters et al. reported suggestive but non-significant associations in small cohorts, while Biobank-based analysis revealed polygenic overlap with psychiatric traits but no genome-wide significant loci [9–11]. These findings point to a modest SNP-based heritability and polygenic architecture for fibromyalgia, underscoring the need for multivariate approaches to uncover shared genetic mechanisms with its common psychiatric comorbidities.

To address these challenges, we employed Genomic Structural Equation Modeling (Genomic SEM), an advanced statistical framework designed to model the genetic architecture of multiple phenotypes using genome-wide association study (GWAS) summary statistics simultaneously [12]. Unlike traditional pairwise methods, such as linkage disequilibrium score regression (LDSC) [13], Genomic SEM identifies latent genetic factors that capture shared genetic variance across multiple traits, thereby enhancing statistical power and the precision of detecting shared genetic influences. While Multi-Trait Analysis of GWAS (MTAG) also leverages data from multiple traits to increase power, it primarily focuses on the discovery of genetic variants without explicitly modeling the underlying shared genetic structure [14]. In contrast, Genomic SEM not only enhances detection power but also provides insights into the latent genetic architecture, facilitating a deeper understanding of the biological mechanisms contributing to the co-occurrence of these complex traits [12].

In this study, we utilized Genomic SEM to explore the shared genetic architecture of fibromyalgia, insomnia, depression, and anxiety. By identifying a common genetic factor, we aimed to elucidate candidate causal variants, associated genes, and relevant biological pathways. Additionally, we examined modifiable risk factors and explored brain structural and functional changes linked to this genetic component. Furthermore, we identified potential blood biomarkers and therapeutic targets, offering valuable insights into the biological mechanisms driving the comorbidity of these conditions, aiming to provide a comprehensive understanding of the shared genetic underpinnings of fibromyalgia and its associated psychiatric disorders.

## Methods

### Ethics approval

All GWASs included in this study had been approved by a relevant review board. No additional ethics approval was required as this study was conducted based on summary-level statistics.

### Study design

The study comprised several key steps: [1] a phenotype-specific GWAS meta-analysis was conducted on the phenotypes of fibromyalgia, insomnia, depression, and anxiety, respectively, to increase statistical power; [2] Genomic SEM was employed to identify the common genetic factor underlying these conditions; [3] downstream analyses were performed to pinpoint lead single nucleotide polymorphisms (SNPs), as well as the most likely implicated genes, pathways,

and tissues associated with this common factor; [4] the Latent Causal Variable (LCV) method was applied to identify modifiable risk factors for the common factor and to determine which diseases might be influenced by it; [5] brain-wide Mendelian randomization (MR) was utilized to identify brain regions linked to the common factor; [6] proteome-wide MR was conducted to identify potential blood biomarkers, and [7] multi-layer brain molecular quantitative trait locus (QTL) analyses was conducted to investigate the underlying mechanisms of mvFibroPsych. The flowchart of the study design is presented in Fig 1.

### Genome-wide association meta-analysis

We conducted phenotype-specific GWAS meta-analyses for fibromyalgia, insomnia, depression, and anxiety, respectively. GWAS data for fibromyalgia were sourced from the BioMe Biobank [11], while data for insomnia and depression were obtained from the UK Biobank [15,16], and anxiety data from the iPSYCH consortium [17]. Additionally, corresponding GWAS datasets for these phenotypes were retrieved from FinnGen R11 to further strengthen the analysis [18]. Comprehensive quality control procedures were implemented across all datasets, including pre-imputation filtering, imputation, and post-imputation assessments, with adjustments for age, sex, and principal components to control for population stratification. All four phenotypes—fibromyalgia, insomnia, depression, and anxiety—were modeled as binary (case–control) traits in

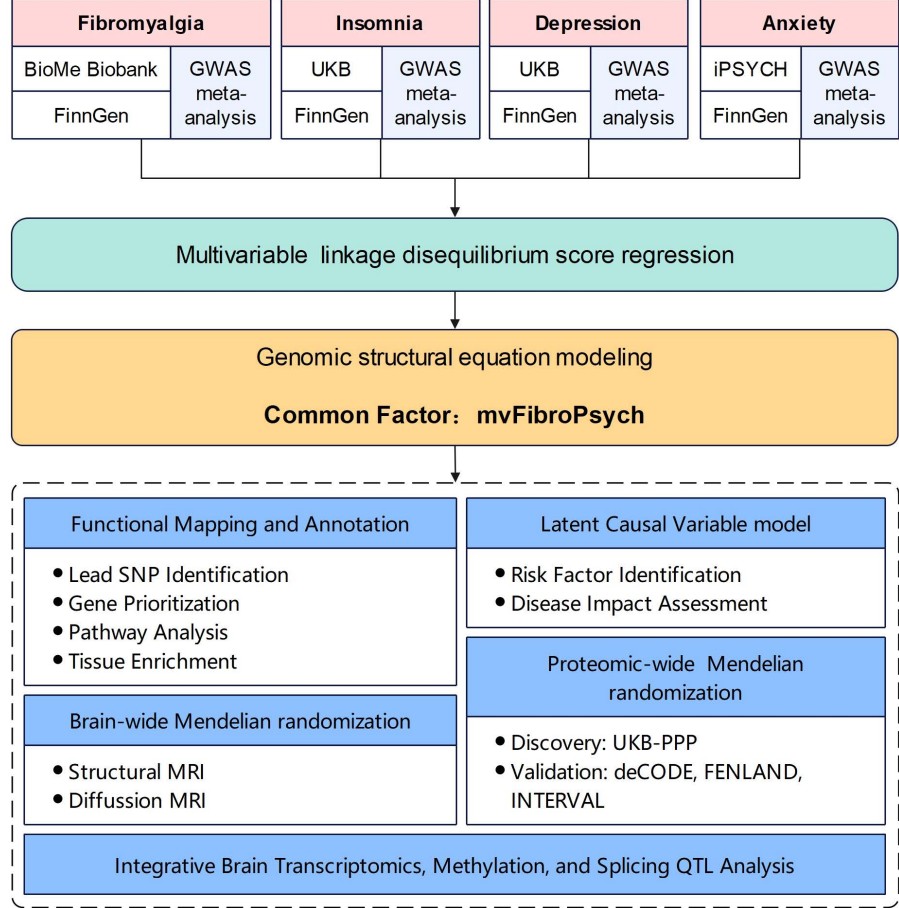

**Fig 1. Study overview.** A flow chart of study design. GWAS, Genome-wide association study; UKB, UK Biobank; SNP, Single-Nucleotide Polymorphism; MRI, Magnetic resonance imaging; QTL, Quantitative trait locus.

the meta-analysis. Fibromyalgia cases were defined based on the ICD-10 code M79.7. Insomnia, depression, and anxiety were defined according to cohort-specific criteria, including self-reported symptoms, clinical interviews, or diagnostic codes, as described in the original GWAS publications from which the summary statistics were obtained. Relevant references and sample size details are provided in S1 Table. We computed the effective sample size (Neff) for each cohort using the standard case–control formula [13]. For meta-analyses of multiple cohorts, we computed Neff for each component dataset separately and summed them to generate the total effective sample size per phenotype. The meta-analysis was performed using the fixed-effect inverse-variance-weighted approach in METAL [19], ensuring robust and reliable combined estimates across studies. All subsequent analyses were conducted based on the summary statistics from this meta-analysis.

## Genetic common factor underlying fibromyalgia, insomnia, depression, and anxiety

The GWAS of the latent common factor for fibromyalgia, insomnia, depression, and anxiety (mvFibroPsych) was conducted by jointly modeling the cross-trait liability of these conditions using a common factor model within the framework of Genomic SEM [12]. We selected the common factor model based on prior evidence of substantial genetic correlations among the included traits, and in line with prior GenomicSEM studies that have modeled shared latent genetic liabilities [12,20–22]. The LDSC was undertaken to investigate the genetic correlations across the four traits. The Genomic SEM integrates genetic correlations and SNP heritability derived from GWAS summary statistics of individual traits, even when the sample overlap is variable or unknown. This method enables the modeling of multivariate genetic associations across phenotypes and can identify variants that influence shared genetic liability across multiple traits [12]. Prior to the multivariable GWAS, allele alignment across fibromyalgia, insomnia, depression, and anxiety was performed using the HAPMAP3 reference panel. Quality control was applied by selecting SNPs with a minor allele frequency greater than 0.01 and an INFO score above 0.9. The multivariable LDSC was then estimated across fibromyalgia, insomnia, depression, and anxiety using the 1000G European reference panel.

The multivariate GWAS was implemented using the commonfactorGWAS() function in GenomicSEM. The latent factor model was estimated using diagonally weighted least squares (DWLS). The regression is performed at the summary-statistics level, and no additional covariates are included at this stage, as covariate adjustment (e.g., age, sex, PCs) had already been applied in the original GWAS of each trait. The analysis was restricted to individuals of European ancestry to ensure LD homogeneity. The effective sample size for the mvFibroPsych GWAS was calculated following the GenomicSEM recommendations. Finally, we integrated the LDSC output with the fibromyalgia, insomnia, depression, and anxiety summary statistics to perform the multivariable common factor GWAS. The GWAS output included SNP-level estimates of beta, standard error, Z-score, and two-sided P-value for the association with the latent factor. In addition, we computed $Q_{SNP}$ statistics and associated $Q_{p-values}$ to test for heterogeneity in SNP effects across the contributing traits. The $Q_{SNP}$ statistic evaluates whether the effect of a SNP is fully mediated by the common factor or whether trait-specific residual effects remain. SNPs with significant $Q_{SNP}$ values indicate potential heterogeneity and were reported separately.

To evaluate model stability and trait-specific contribution, we performed sensitivity analyses using multiple combinations of traits. These included (i) a three-trait model excluding fibromyalgia, (ii) three-trait models pairing fibromyalgia with two psychiatric traits, and (iii) user-specified pairwise models with factor loading of fibromyalgia was fixed to 1. Model fit and factor loading significance were used to assess robustness and overlap structure. All models were run using the GenomicSEM v0.0.5 R package with default settings. Model fit was evaluated using Comparative Fit Index (CFI) and Standardized Root Mean Square Residual (SRMR). All multivariate GWAS results were independently replicated by multiple members of the research team using separate computational pipelines and software installations.

## Genome-wide association study annotation

The post-GWAS analysis of the mvFibroPsych was conducted using the Functional Mapping and Annotation (FUMA) platform (https://fuma.ctglab.nl/home) to perform SNP annotation, gene prioritization, and enrichment analyses [23]. SNPs

were annotated based on their functional consequences using reference databases, including Annotate Variation (ANNO-VAR) and The Combined Annotation-Dependent Depletion (CADD) scores, to identify likely causal variants [24,25]. Gene prioritization was carried out by mapping significant SNPs to genes through positional mapping, expression quantitative trait loci (eQTL) mapping, and chromatin interaction data. Additionally, gene-based association analysis was performed using the Multi-marker Analysis of Genomic Annotation (MAGMA) to identify genes associated with the mvFibroPsych. Tissue enrichment and pathway enrichment analyses were then undertaken to determine the biological pathways and tissues most relevant to the identified genes, utilizing databases such as GTEx for tissue-specific expression and MsigDB for pathway annotations. Standard parameters and thresholds were applied throughout these analyses to ensure the robustness of the results.

### Identification of the modifiable risk factors for mvFibroPsych

The LCV method was applied to infer causal relationships between the common factor and other traits phenome-wide. Given its ability to leverage genome-wide genetic correlation patterns without requiring strong instruments, LCV was specifically selected for identifying potentially causal directions between mvFibroPsych and complex traits, especially where effect size estimation was not the primary aim [26]. The LCV model introduces a latent variable (L) that is assumed to exert causal effects on both traits through their genetic correlation. This model evaluates whether one trait potentially causes the other (vertical pleiotropy) by comparing the strength of association between L and each trait. An absolute genetic causality proportion (GCP) value exceeding 0.6 and a p-value below the FDR-corrected (Benjamini–Hochberg false discovery rate) threshold is considered evidence of a causal link between the two traits. A positive GCP suggests that trait A is likely to influence trait B, while a negative GCP implies the reverse [26]. The phenome-wide causal relationships between the common genetic factor identified in our analysis and various traits were examined using GWAS data from the UK Biobank, specifically the second wave of data released by the Neale Lab (www.nealelab.is/uk-biobank/), which primarily includes individuals of European ancestry. The covariates adjusted in the UK Biobank GWAS data have been extensively described elsewhere [27].

### Brain-Wide Mendelian randomization analyses for the relationship between Brain imaging-derived phenotypes and mvFibroPsych

The two-sample Mendelian randomization framework was employed to quantify the direction and magnitude of putative causal effects between Brain imaging-derived phenotypes (IDPs) and mvFibroPsych. GWAS summary statistics for IDPs were obtained from a cohort of 33,224 individuals of European ancestry, as released in the 2020 UK Biobank dataset [28,29]. IDPs were obtained from structural MRI (sMRI) and diffusion MRI (dMRI) modalities, with sMRI capturing brain anatomical variations and dMRI assessing structural connectivity between brain regions. We filtered the initial 3,935 IDPs using a stepwise approach to ensure reliable and reproducible results, following the methodology by Yang et al [30]. First, we addressed redundancy by removing 1,201 IDPs measured in the same brain regions with different tools, retaining only those assessed by commonly used methods such as FreeSurfer, FIRST, tract-based spatial statistics, and probabilistic tractography. Next, we excluded 358 IDPs derived from small brain areas with low contrast in MR images. After applying these filters, we selected 227 sMRI and 360 dMRI IDPs for further analysis.

For assessing the impact of IDPs on mvFibroPsych, we applied a more lenient set of parameters for instrumental variable (IV) selection (p-value threshold of $5 \times 10^{-6}$, an $r^2$ threshold of 0.001, and a 1 Mb window size). This approach was necessitated by the relatively smaller sample size of the IDPs, which limited the number of SNPs reaching the conventional genome-wide significance threshold of $5 \times 10^{-8}$. Conversely, for evaluating the influence of mvFibroPsych on IDPs, we utilized more stringent IV selection criteria (p-value threshold of $5 \times 10^{-8}$, an $r^2$ threshold of 0.001, and a 1 Mb window size) to ensure robust and reliable findings. LD estimation was based on the 1000 Genomes European data (phase 3). Outlier IVs identified by RadialMR were excluded from the analysis, and any IVs with associations to potential

confounders ($p < 5 \times 10^{-8}$ based on the GWAS Catalog) were also removed. The primary analysis was conducted using the Inverse-Variance Weighted (IVW) method [31]. The MRPRESSO package was employed to address potential horizontal pleiotropy [32]. Sensitivity analyses were performed to evaluate the robustness of the results [33]. Additionally, the MR Steiger method was utilized to assess the true direction of causality [34].

## Proteome-Wide Mendelian Randomization Analysis for Biomarkers and Therapeutic Targets for mvFibroPsych

The Summary data-based Mendelian Randomization (SMR) method was employed to investigate the relationship between blood protein levels and mvFibroPsych, to identify potential blood biomarkers [35]. This analysis utilized cis-pQTLs identified from the UK Biobank Pharma Proteomics Project (UKB-PPP), which included plasma samples from 54,219 individuals of European ancestry and measured 2,923 plasma proteins using the Olink Explore platform [36]. Cis-pQTLs were defined as SNPs located within 1 Mb of the transcription start site (TSS) of the gene encoding the corresponding protein. Only index cis-pQTLs associated with protein levels at a genome-wide significance threshold ($P < 5 \times 10^{-8}$) were included in the SMR analysis. To differentiate between true pleiotropy and linkage, where distinct but linked causal variants could influence protein levels and the disease phenotype, the Heterogeneity in Dependent Instruments (HEIDI) test was conducted. A protein with a p-SMR below the FDR correction threshold and a p-HEIDI exceeding 0.01 was considered to have a true causal relationship with mvFibroPsych, not driven by linkage disequilibrium. Additionally, alternative datasets from the deCODE (N = 35,559), the FENLAND (N = 10,708), and the INTERVAL (N = 3301) datasets were applied as the validation cohort for this analysis [37].

## Integrative analyses of gene expression, DNA methylation, and alternative splicing for mvFibroPsych

To investigate the influence of gene expression, DNA methylation, and alternative splicing on mvFibroPsych, we applied the SMR method, incorporating eQTL (estimated effective N = 2,443), mQTL (estimated effective N = 1160), and sQTL (estimated effective N = 2,443) data from the BrainMeta data sources [38,39]. First, we used cis-eQTLs to assess the impact of brain-wide gene expressions on mvFibroPsych, where cis-eQTLs were defined as SNPs located within 1 Mb of the TSS of the corresponding gene and significantly associated with gene expression at the genome-wide significance threshold ($P < 5 \times 10^{-8}$). Subsequently, mQTL and sQTL data were utilized to explore whether DNA methylation or alternative splicing of these genes also exhibited causal effects on mvFibroPsych. The SMR method was used to estimate the causal relationship between these molecular traits and mvFibroPsych, and the HEIDI test was performed to distinguish true pleiotropy from linkage. A significant SMR result (p-SMR below the FDR correction threshold) combined with a HEIDI p-value exceeding 0.01 indicated a likely causal association between the molecular trait and mvFibroPsych, free from confounding by linkage disequilibrium.

## Results

### Latent common factor GWAS estimation

After meta-analysis, the estimated effective sample sizes (Neff) were 17,827 for fibromyalgia, 115,173 for insomnia, 391,264 for depression, and 191,496 for anxiety disorder (S1 Table). In particular, meta-analysis modestly increased the number of genome-wide significant loci, supporting improved statistical power. A positive genetic correlation was observed among the four univariate input GWASs with $r_g$ ranging from 0.55 to 0.84, (P-value ranging from $1.23 \times 10^{-94}$ to $1.29 \times 10^{-10}$) (Fig 2A and S2 Table). Model fit for the common factor shows a good fit to the implied genetic covariance matrix between fibromyalgia, insomnia, depression, and anxiety with a comparative fit index (CFI) = 0.999 and a standardized root mean square residual (SRMR) = 0.015, providing evidence for a shared genetic factor. The standard factor loadings for the four input phenotypes are presented in Fig 2B, with factor loading varying from 0.63 to 0.94, aligned well with prior literature [20].

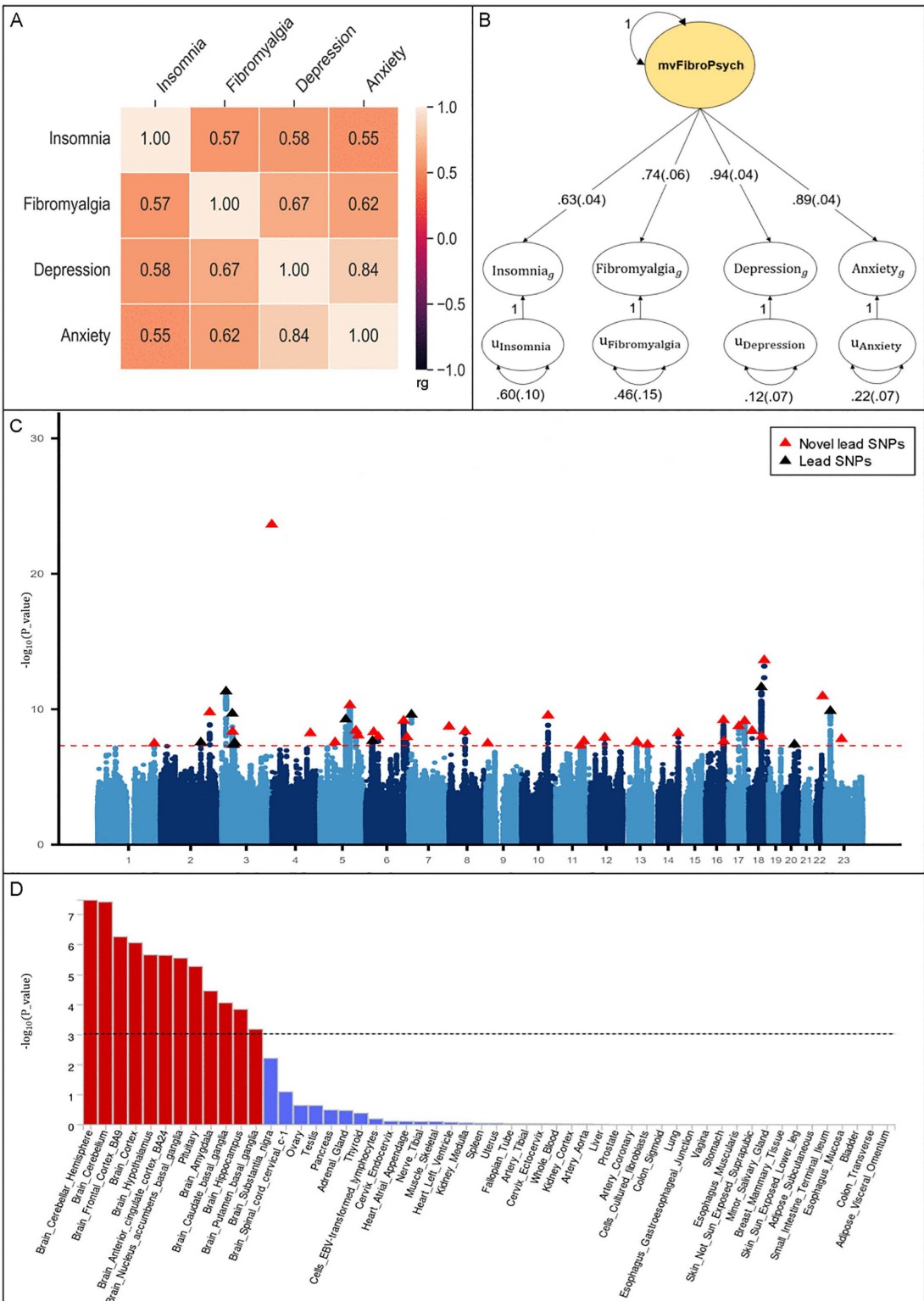

**Fig 2. Multivariate GWAS modeled with Genomic SEM. A.** Genetic correlations across the four univariate phenotypes by using pairwise LDSC; **B.** Path diagram of the common factor model estimated with Genomic SEM, with standardized factor loadings (standard error in parentheses); **C.** Manhattan plot showing SNP associations (−log10(P-value)) with mvFibroPsych, ordered by chromosome. The red dashed line indicates the threshold for

conventional genome-wide significance (P-value = 5 × 10⁻⁸). P-values are derived from two-sided Wald tests for each SNP on mvFibroPsych; **D.** Tissue enrichment from MAGMA analysis by using GTEx V8. The black dashed line indicates the threshold for the FDR-corrected P-value. GWAS, Genome-wide association study; SEM, Structural Equation Modeling; SNP, Single-Nucleotide Polymorphism.

In the sensitivity analyses, we systematically assessed the robustness of the latent factor model by constructing GenomicSEM models using subsets of the original four phenotypes. Across all three-trait configurations—including fibromyalgia with any two psychiatric traits—the models demonstrated consistently acceptable fit (CFI > 0.95, SRMR < 0.08) (S3 Table), and fibromyalgia retained statistically significant factor loadings, indicating its stable contribution to the shared genetic architecture. In pairwise models where the factor loading of fibromyalgia was fixed to 1 and the loading of the psychiatric trait was freely estimated, the psychiatric traits also exhibited statistically significant loadings onto the latent factor (S3 Table). These findings confirm that the observed multivariate structure is not solely driven by highly powered psychiatric traits, and that fibromyalgia contributes non-redundant shared variance, despite its smaller sample size relative to the other phenotypes.

The common factor model built in the Genomic SEM was applied to incorporate individual variants, enabling the generation of a multivariate GWAS that estimated 7,627,423 associations at the SNP level for the shared factor mvFibroPsych. A total of 49 lead SNPs across 43 independent loci were identified at genome-wide significance (P < 5 × 10⁻⁸) (Fig 2C and S4 Table). Among these, 32 loci had not been previously reported in any of the eight input GWAS datasets, underscoring the enhanced power and resolution offered by the GenomicSEM approach. To assess the relative contribution of each trait to the genome-wide significant loci, we examined the Z-statistics of lead SNPs across the four input phenotypes. The results revealed that the strongest effects were not consistently driven by depression; in many cases, fibromyalgia, insomnia and anxiety had comparable or greater Z-values (S5 Table). This indicates that the top SNPs identified in the mvFibroPsych GWAS capture shared variance across multiple traits, rather than being solely attributable to the best-powered phenotype. To evaluate whether genome-wide significant loci from the mvFibroPsych GWAS act homogeneously across all component phenotypes, we computed Cochran's Q statistic for each SNP. Among the significant loci, only one SNP showed evidence of cross-trait heterogeneity (rs1245129, Q_P-value less than the Bonferroni corrected threshold), suggesting trait-specific effects at this locus. Notably, several genome-wide significant loci for the mvFibroPsych latent factor showed modest or non-significant marginal effects in the individual univariate GWASs. For example, the lead variant rs4865477 reached genome-wide significance in the latent factor GWAS (P = 2.31 × 10⁻²⁴) despite all input Z-scores for the individual traits falling within ±1.96. Importantly, rs4865477 did not show significant Q statistics, indicating that its effect is consistent with a pure common-factor model and reflects shared genetic liability across traits rather than trait-specific heterogeneity. Full results, including Q statistics and heterogeneity-adjusted p-values, are provided in S5 Table. The genomic control ($\lambda_{GC}$) was estimated at 1.207, with an LDSC intercept of 0.812 (se = 0.0074). The effective sample size for the mvFibroPsych GWAS was estimated to be 5,927,502. To evaluate whether the observed inflation in test statistics reflected true polygenicity or residual confounding, we calculated the attenuation ratio using the formula (LDSC intercept − 1)/ ($\lambda_{GC}$ − 1), which yielded an attenuation ratio of approximately -0.908. This negative value is atypical in single cohort GWAS but has been observed in multivariate analyses where phenotype sparsity, cohort heterogeneity, or conservative correction procedures may drive intercept deflation [40–42]. We also confirmed that the heritability Z-scores and LDSC cross-trait intercepts showed no evidence of uncontrolled population structure.

## FUMA annotation

The SNP-to-gene mapping was performed using FUMA (v1.6.1). Among all SNPs in LD (r² ≥ 0.6) with independent genome-wide significant SNPs, 54.9% were mapped to genes by positional proximity, 48.1% via eQTL mapping, and 0% via chromatin interaction. Mapping strategies were applied in parallel, and some SNPs were annotated by multiple methods (S6 Table). Based on position mapping and eQTL mapping, gene prioritization identified 342 protein-coding genes

associated with mvFibroPsych (S7 Table). The MAGMA gene-set analyses revealed that mvFibroPsych has an enrichment in the pathways involved in synaptic function (S8 Table). The MAGMA tissue expression analysis using the GTEx V8 53 general tissue datasets shows that the brain and the pituitary are the most associated tissues (Fig 2D).

## Modifiable risk factors with mvFibroPsych

Of the 1,266 phenotypes examined by LCV, a total of 133 phenotypes demonstrated a |GCP| greater than 0.6 and remained significant after FDR correction. Among these, 131 phenotypes showed evidence of genetically causal effects on mvFibroPsych (negative GCP), while mvFibroPsych itself was identified as a potential causal factor for 2 phenotypes (positive GCP) (Fig 3 and S9 Table). The mvFibroPsych showed a positive causal association with "Substances taken for anxiety: Drugs or alcohol (more than once)" ($r_g$ = 0.855, GCP = 0.889, FDR = 3.96 × 10$^{-15}$). This strong positive genetic correlation indicates that individuals with higher genetic liability captured by mvFibroPsych are more likely to engage in substance use as a coping mechanism for anxiety, aligning with the affective dysregulation underlying the latent construct. Positive causal associations were also observed in diseases of "Back pain for 3+ months" ($r_g$ = 0.385, GCP = 0.749, FDR = 5.74 × 10$^{-3}$). Traits with GCP values below –0.6 showed strong evidence of exerting upstream causal influences

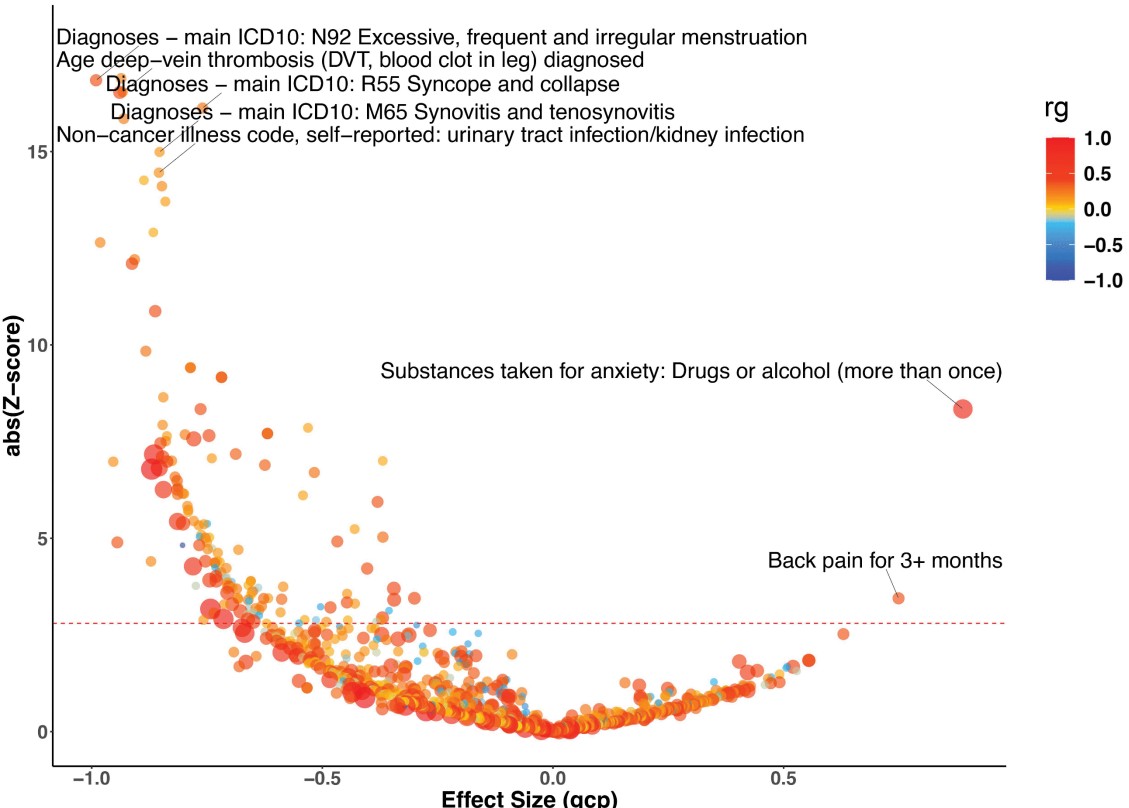

**Fig 3. Potentially causal associations for mvFibroPsych.** Causal architecture plots illustrating the latent causal variable exposome-wide analysis results. Each dot represents a trait with a significant genetic correlation with mvFibroPsych. The y-axis shows the genetic causality proportion (GCP) absolute Z-score (statistical significance), whilst the x-axis exhibits the GCP estimate. The red dashed lines represent the statistical significance threshold (FDR < 0.05), while the division for traits causally influencing mvFibroPsych (on the left) and traits causally influenced by mvFibroPsych (on the right) is represented by the grey dashed lines. Phenotypes in blue show a negative genetic correlation with mvFibroPsych, while phenotypes in red show a positive genetic correlation with mvFibroPsych.

on mvFibroPsych. Notably, excessive, frequent and irregular menstruation (ICD10: N92), deep vein thrombosis (DVT), syncope and collapse (ICD10: R55), synovitis and tenosynovitis (ICD10: M65), and urinary tract/kidney infection demonstrated pronounced effects, suggesting that these disorders may act as upstream contributors predisposing individuals to the latent mvFibroPsych factor ($r_g > 0$, GCP < -0.6, FDR < 0.05) (Fig 3 and S9 Table).

### Brain-Wide Mendelian Randomization Analyses for Relationship Between Brain and mvFibroPsych

The F-statistics for the selected IVs were all greater than 10, indicating a low likelihood of weak instrument bias. In the analysis of the causal direction from brain imaging-derived phenotypes (IDPs) to mvFibroPsych, 11 out of 227 sMRI phenotypes and 14 out of 360 dMRI phenotypes demonstrated a suggestive causal association with mvFibroPsych (p < 0.05) (S10 and S11 Tables and Fig 4). However, after FDR correction, only one dMRI phenotype—fractional anisotropy (FA) in the splenium of the corpus callosum—remained significantly associated with mvFibroPsych. Specifically, per 10 standard deviation (SD) increase in FA in the splenium of the corpus callosum was associated with an 8.1% reduction in the odds of mvFibroPsych (OR = 0.919, 95% CI 0.880–0.960, FDR = 0.048). FA measures the degree of anisotropic diffusion —that is, the extent to which water molecules diffuse more readily along the direction of white matter fibers than perpendicular to them. FA values typically decrease in structurally compromised tissue, such as damaged or demyelinated white matter tracts. The observed negative causal association between FA in the splenium of the corpus callosum and mvFibroPsych suggests that the integrity of this brain region may play a protective role against the development of mvFibroPsych. The MR Steiger test confirmed the directionality of the causal relationship. Furthermore, sensitivity analyses, including MR-Egger, MRPRESSO, and leave-one-out tests, demonstrated the robustness of the MR estimates.

No significant associations were found in the bidirectional Mendelian Randomization analysis examining the potential causal effects of mvFibroPsych on brain IDPs (S12 and S13 Tables). This lack of evidence suggests that brain changes are more likely to be a causal factor in mvFibroPsych rather than a consequence of the condition, supporting the hypothesis that neuroanatomical alterations may predispose individuals to mvFibroPsych rather than result from it.

### Proteome-Wide Mendelian Randomization Analysis for Biomarkers and Therapeutic Targets for mvFibroPsych

Of the 2,923 proteins examined in the UKB-PPP datasets, 1,987 had sufficient instrumental variables to yield reliable effect estimates, and 25 proteins met the FDR-corrected significance threshold (P < 5.29 × 10⁻⁴, FDR < 0.05). Among these, five proteins—*UBE2L6*, *VWC2*, *BTN3A2*, *FES*, and *HSPA1A*—exhibited a P_HEIDI value greater than 0.01, suggesting that their association with mvFibroPsych is not due to linkage disequilibrium but rather reflects a true causal relationship (S14 Table). Notably, all five proteins showed a positive causal association with mvFibroPsych, underscoring their potential relevance as biomarkers for this condition. Remarkably, *VWC2* demonstrated a significant positive association with the occurrence of mvFibroPsych, consistently passing the FDR correction threshold in the UKB-PPP and all the validation datasets including the deCODE, the FENLAND, and the INTERVAL datasets (Fig 5 and S15-17 Tables). The effect directions were consistent, and the P_HEIDI value exceeded 0.01, indicating a robust causal relationship, prioritizing the potential importance of *VWC2* as a biomarker for mvFibroPsych.

### Multi-layer Brain QTL Analyses for mvFibroPsych

Using the SMR-HEIDI method, we identified 20 genes whose brain-wide expression had significant associations with mvFibroPsych (FDR < 0.05, p-HEIDI > 0.01) (S18 Table). Among these, the top three genes were *SLC12A5* (beta = 0.0044, FDR = 1.83 × 10⁻³, p-HEIDI > 0.01), BPTF (beta = -0.0248, FDR = 6.45 × 10⁻³, p-HEIDI > 0.01), and FURIN(beta = -0.0039, FDR = 7.78 × 10⁻³, p-HEIDI > 0.01), indicating strong and statistically significant associations between their expression levels and mvFibroPsych. The methylation QTL (mQTL) analysis further identified four genes—*AMZ1* (cg23627948, beta = 0.001, FDR = 2.1 × 10⁻², p-HEIDI > 0.01), *CD40* (cg09053081, beta = 0.0048, FDR = 2.2 × 10⁻², p-HEIDI > 0.01),

| MRI | Exposure | IVs | | OR (95% CI) | P-value |
|---|---|---|---|---|---|
| Structural | aparc-Desikan lh area superiortemporal | 35 | | 0.912 (0.865, 0.963) | 0.001* |
| | aparc-Desikan lh thickness transversetemporal | 16 | | 0.916 (0.862, 0.974) | 0.005* |
| | aparc-Desikan rh volume superiorfrontal | 24 | | 0.904 (0.841, 0.971) | 0.006* |
| | aparc-Desikan rh area superiorfrontal | 45 | | 0.910 (0.850, 0.975) | 0.007* |
| | aparc-Desikan rh thickness paracentral | 35 | | 0.951 (0.916, 0.987) | 0.008* |
| | aseg global volume WM-hypointensities | 34 | | 1.056 (1.008, 1.106) | 0.021* |
| | IDP T1 FIRST right accumbens volume | 33 | | 1.080 (1.007, 1.158) | 0.031* |
| | IDP T1 FIRST left caudate volume | 74 | | 1.057 (1.004, 1.113) | 0.033* |
| | aparc-Desikan rh thickness parsopercularis | 23 | | 0.923 (0.855, 0.996) | 0.038* |
| | aparc-Desikan rh thickness parahippocampal | 18 | | 0.930 (0.868, 0.997) | 0.042* |
| Diffussion | **IDP dMRI TBSS FA Splenium of corpus callosum** | **53** | | **0.919 (0.880, 0.960)** | **<0.001**\*\* |
| | IDP dMRI ProbtrackX ISOVF ifo l | 51 | | 0.941 (0.901, 0.982) | 0.006* |
| | IDP dMRI TBSS ICVF Posterior limb of internal capsule R | 80 | | 0.948 (0.912, 0.985) | 0.007* |
| | IDP dMRI TBSS ISOVF Superior fronto-occipital fasciculus L | 22 | | 0.924 (0.870, 0.981) | 0.009* |
| | IDP dMRI TBSS FA Uncinate fasciculus R | 30 | | 0.941 (0.897, 0.987) | 0.012* |
| | IDP dMRI TBSS FA Inferior cerebellar peduncle L | 45 | | 0.956 (0.923, 0.990) | 0.012* |
| | IDP dMRI TBSS OD Superior corona radiata R | 46 | | 0.963 (0.934, 0.993) | 0.015* |
| | IDP dMRI TBSS ICVF Superior longitudinal fasciculus L | 81 | | 0.962 (0.932, 0.993) | 0.018* |
| | IDP dMRI TBSS MD Superior longitudinal fasciculus L | 74 | | 1.046 (1.007, 1.087) | 0.021* |
| | IDP dMRI TBSS ISOVF Body of corpus callosum | 44 | | 0.951 (0.908, 0.995) | 0.030* |
| | IDP dMRI TBSS ISOVF Sagittal stratum R | 48 | | 0.949 (0.904, 0.996) | 0.035* |
| | IDP dMRI TBSS ISOVF Retrolenticular part of internal capsule R | 21 | | 0.954 (0.912, 0.998) | 0.039* |
| | IDP dMRI ProbtrackX ICVF atr l | 83 | | 0.964 (0.930, 0.998) | 0.040* |
| | IDP dMRI TBSS ISOVF Anterior corona radiata L | 37 | | 0.951 (0.907, 0.998) | 0.043* |

0.80 0.85 0.90 0.95 1.0 1.05 1.1 1.15 1.2
Odds Ratio

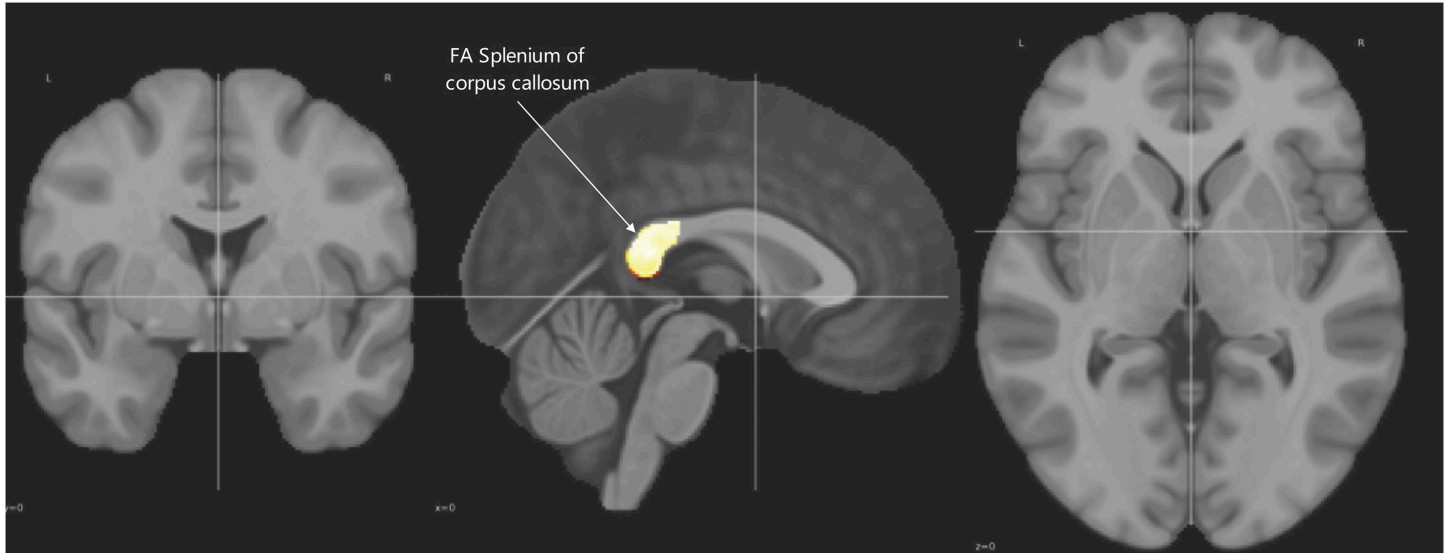

**Fig 4. Suggestive causal associations between brain structural as well as diffusion IDPs and mvFibroPsych.** * P-value<0.05, ** FDR<0.05.

GSDME (cg27436603, beta = -0.0087, FDR = $2.0 \times 10^{-2}$, p-HEIDI > 0.01), and *TMEM258* (cg18171955, beta = -0.0033, FDR = $3.6 \times 10^{-2}$, p-HEIDI > 0.01)—where methylation was significantly associated with mvFibroPsych (S19 Table). In the splicing QTL (sQTL) analysis, five genes—*AREL1*, *ARHGAP19*, *BPTF*, *CD40*, and *OTOA*—exhibited significant splicing variants associated with mvFibroPsych (S20 Table). Notably, *CD40* once again demonstrated significant associations at splicing levels (splicing beta = 0.0103, FDR = $1.8 \times 10^{-2}$, p-HEIDI > 0.01), underscoring its potential role as a key regulatory

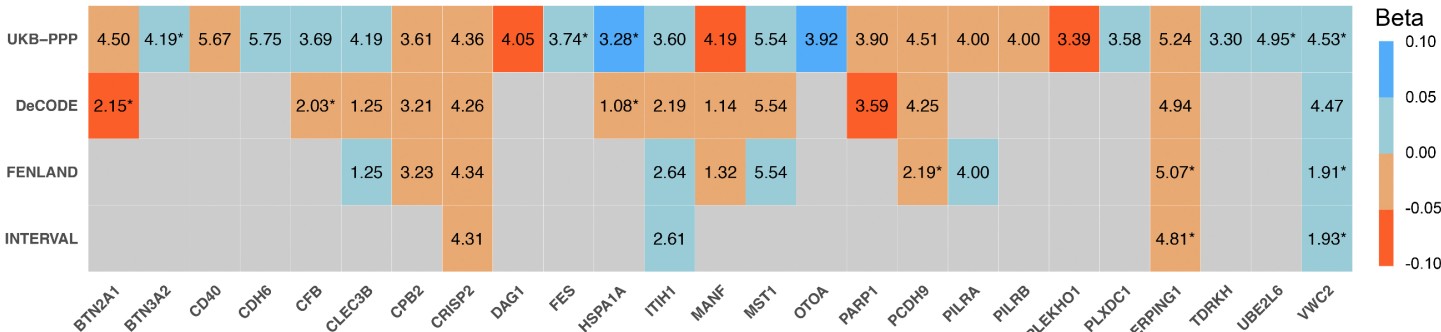

**Fig 5. Significant causal associations between plasma proteins and mvFibroPsych (FDR<0.05).** Proteins in blue show a positive causal association with mvFibroPsych, while phenotypes in red show a negative causal association with mvFibroPsych. * P_HEIDI>0.01.

factor in the pathology of mvFibroPsych (Fig 6). Interestingly, the functional roles of these genes align with known pathways implicated in neuroinflammation, synaptic plasticity, and cellular apoptosis, which are hypothesized to be relevant to the pathogenesis of fibromyalgia and associated psychological conditions. The association of genes such as *BPTF* and *ARHGAP19*, which are involved in chromatin remodeling and cell signaling, respectively, provides further evidence that mvFibroPsych may be driven by dysregulated molecular processes that affect both neurodevelopment and neurodegeneration [43,44].

## Discussion

This study aimed to investigate the shared genetic architecture of fibromyalgia and its associated psychiatric conditions—insomnia, depression, and anxiety—through a comprehensive genomic approach. By utilizing the Genomic SEM and downstream analyses, we identified a common genetic factor underlying these conditions, with potential clinical and biological implications. Our findings contribute to a growing body of evidence that underscores the genetic overlap between fibromyalgia and psychiatric disorders, shedding light on potential pathways, risk factors, and biomarkers that may inform future therapeutic strategies.

The discovery of a common genetic factor with significant factor loadings for fibromyalgia, insomnia, depression, and anxiety supports the hypothesis that these conditions share genetic underpinnings. It is noteworthy that some fibromyalgia-specific loci, including previously reported borderline-significant hits in the FinnGen dataset (rs139332433 and rs56132815), did not replicate in our multivariate model. This likely reflects the fact that mvFibroPsych captures shared, rather than

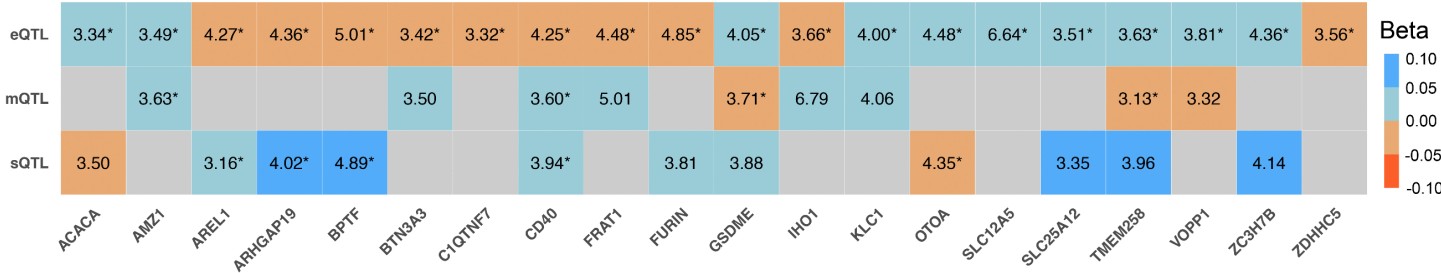

**Fig 6. Significant causal associations between brain gene expression, DNA methylation, and alternative splicing and mvFibroPsych (FDR<0.05).** Proteins in blue show a positive causal association with mvFibroPsych, while phenotypes in red show a negative causal association with mvFibroPsych. The number in each square indicates the threshold for the log-transformed P-value. * P_HEIDI>0.01.

disorder-specific, genetic effects. Notably, although insomnia showed moderate genetic correlations with the other traits, its factor loading was lower than depression and anxiety. This may reflect a more heterogeneous or partly distinct genetic basis for insomnia, including contributions from circadian, metabolic, or non-psychiatric pathways not fully represented in the latent factor [45,46]. Several loci in our study reached genome-wide significance for the mvFibroPsych latent factor despite weak associations in the individual input GWASs and, in some cases, limited surrounding linkage disequilibrium support in the mvFibroPsych Manhattan plots. This pattern reflects a fundamental distinction between latent factor GWAS and conventional single-phenotype GWAS. While traditional GWAS tests associations with individual traits, GenomicSEM evaluates SNP effects on a multivariate liability dimension defined by the genetic covariance structure across correlated phenotypes. By integrating small but directionally consistent pleiotropic effects across traits, GenomicSEM can identify loci that are not detectable in any single-trait GWAS and may not necessarily exhibit classical LD-supported clustering [12]. In addition, the heterogeneity analysis revealed that one genome-wide significant locus (rs1245129) exhibit non-uniform effects across the component phenotypes, indicating potential trait-specific influences. This highlights the importance of interpreting multivariate GWAS results in light of both shared and trait-specific architectures. The pathways identified—specifically those related to synaptic function—suggest that neuronal signaling mechanisms may play a critical role in the shared etiology of these conditions [47,48]. Moreover, the identification of the pituitary and brain tissues as key tissues associated with mvFibroPsych suggests that neuroendocrine dysregulation may contribute to this shared genetic liability, further implicating the hypothalamic-pituitary-adrenal (HPA) axis in the comorbidity between fibromyalgia and psychiatric conditions.

The phenome-wide LCV analyses further elucidate the multidimensional nature of the mvFibroPsych factor. Its strong positive causal associations with the phenotypes "Substances taken for anxiety: Drugs or alcohol (more than once)" and "Back pain for 3+ months" suggest that individuals with a higher genetic liability captured by mvFibroPsych are more likely to engage in substance use and to experience persistent pain symptoms. These behaviors and clinical manifestations may reflect maladaptive coping and heightened somatic sensitivity arising from shared affective dysregulation [49]. This finding aligns with the internalizing-spectrum framework and supports the interpretation that mvFibroPsych represents a broad latent dimension of emotional and somatic vulnerability, rather than a construct specific to fibromyalgia. Conversely, traits such as excessive or irregular menstruation (ICD10: N92), deep vein thrombosis, syncope, synovitis, and urinary tract infections exhibited GCP values below –0.6, indicating potential upstream causal influences on mvFibroPsych. These pain-, inflammation-, and circulation-related conditions may act as chronic somatic stressors predisposing individuals to internalizing vulnerability through sustained neuroimmune and neuroendocrine activation. Collectively, these results support a biopsychosomatic framework in which chronic somatic burden and affective dysregulation converge on shared neurobiological mechanisms, shaping the latent liability captured by mvFibroPsych [50]. Our brain-wide MR analysis provided compelling evidence for the role of brain structure in the genetic architecture of mvFibroPsych. Specifically, we identified a negative causal association between the FA in the splenium of the corpus callosum and mvFibroPsych. The corpus callosum has been implicated in several neuropsychiatric disorders, and reduced FA often signifies compromised white matter integrity [51,52]. These findings align with previous studies suggesting that structural brain abnormalities, particularly in white matter, may predispose individuals to neuropsychiatric conditions [53,54]. Importantly, our results suggest that brain changes may be causal in the development of mvFibroPsych, rather than a consequence of the condition. This highlights the potential of neuroimaging as a diagnostic tool for This highlights the potential of neuroimaging as a diagnostic tool for detecting shared neurobiological mechanisms underlying the internalising vulnerability represented by this latent dimension.

Proteome-wide MR analyses identified five proteins—*UBE2L6*, *VWC2*, *BTN3A2*, *FES*, and *HSPA1A*—as causally associated with mvFibroPsych. Of particular interest is *VWC2* (Von Willebrand factor C domain-containing protein 2), which demonstrated consistent effects across all four datasets, underscoring its potential as a biomarker for mvFibroPsych. *VWC2* has been linked to neuroinflammation and extracellular matrix (ECM) regulation, crucial for maintaining synaptic integrity and plasticity [55]. By influencing ECM remodeling, *VWC2* may affect neuronal connectivity, contributing to fibromyalgia and psychiatric

symptoms [56]. The identification of blood biomarkers such as *VWC2* provides a non-invasive avenue for diagnosing and monitoring mvFibroPsych, offering potential clinical applications in early detection and personalized treatment strategies.

The multi-layer QTL analysis uncovered key molecular signatures that may contribute to the pathophysiology of mvFibroPsych. Regarding gene expression levels, the eQTL analysis revealed three key genes—*SLC12A5*, *BPTF*, and *FURIN*—with significant associations with mvFibroPsych. Higher expression of *SLC12A5* (Solute Carrier Family 12 Member 5) is associated with an increased risk of mvFibroPsych. This is consistent with its known role in encoding *KCC2* (Potassium-Chloride Cotransporter 2), a neuron-specific potassium-chloride cotransporter crucial for maintaining inhibitory GABAergic signaling. Upregulation of *SLC12A5* could lead to impaired chloride homeostasis, contributing to increased neuronal excitability and heightened sensitivity to pain, both of which are characteristic of chronic pain conditions like mvFibroPsych [57]. Conversely, *BPTF* (Bromodomain *PHD* Finger Transcription Factor) showed a negative association with the risk of mvFibroPsych. *BPTF* is a key subunit of the *NURF* (Nucleosome Remodeling Factor) chromatin remodeling complex, involved in regulating gene expression via chromatin accessibility. Reduced *BPTF* expression could impair chromatin remodeling, possibly contributing to abnormal gene expression patterns in neural cells, which might exacerbate the neurodevelopmental and neuroinflammatory processes hypothesized to drive mvFibroPsych [58]. Therefore, *BPTF* downregulation could be protective by maintaining proper chromatin dynamics, preventing maladaptive neural plasticity changes. Similarly, *FURIN* also exhibited a negative association with mvFibroPsych. *FURIN* is a proprotein convertase involved in processing several precursor proteins, including those implicated in immune regulation and neuroinflammation [59]. Given the proposed role of neuroinflammation in chronic pain and associated psychological disorders, higher *FURIN* expression may help mitigate inflammatory responses, potentially acting as a buffer against the progression of mvFibroPsych pathology. Notably, the consistent involvement of *CD40* across the gene expression levels, DNA methylation, and splicing QTL layers strongly implicates this gene in the molecular etiology of mvFibroPsych. *CD40*, known for its role in immune system regulation, particularly in the context of neuroinflammation, has been widely studied in autoimmune and neurodegenerative diseases [60]. Here, we observed that increased methylation at cg09053081 is positively associated with mvFibroPsych risk, suggesting that altered methylation patterns may contribute to dysregulated *CD40* expression, leading to maladaptive immune responses in the brain. This aligns with the growing body of evidence linking aberrant immune signaling and chronic pain [61]. Interestingly, while *CD40* expression is negatively associated with mvFibroPsych risk, its splicing variations appear to have a positive association, indicating that alternative splicing may produce functionally distinct isoforms that differentially modulate immune responses [62,63]. These findings highlight the complex, multi-layered regulation of *CD40* and its potential dual role in both protective and pathogenic pathways in mvFibroPsych.

While this study provides robust insights into the shared genetic basis of fibromyalgia and its psychiatric comorbidities, certain limitations should be acknowledged. First, the study predominantly included individuals of European ancestry, limiting the generalizability of the findings to other populations. Future research should aim to replicate these findings in diverse populations to assess their applicability across different genetic backgrounds. Second, while large-scale meta-GWAS resources (e.g., ISGC, PGC) exist for insomnia, depression, and anxiety, we did not include these datasets due to limited public access to full SNP-level summary statistics, particularly those involving 23andMe and other restricted-access cohorts. Additionally, these consortia partially overlap with the cohorts we used (e.g., UK Biobank, iPSYCH), which would pose challenges for modeling genetic covariance accurately within the GenomicSEM framework. To avoid bias due to sample overlap and maximize transparency and reproducibility, we prioritized publicly available GWAS datasets with documented non-overlapping samples and harmonized ancestry and quality control procedures. Third, while we included FinnGen data in our GWAS meta-analyses to improve power—particularly for fibromyalgia, which has relatively small case numbers—we acknowledge that this precluded its use as a fully independent replication cohort. As more large-scale biobank datasets (e.g., All of Us, Estonian Biobank) release relevant phenotype-specific summary statistics, future studies may leverage these resources to independently validate and generalize the current findings. Fourth, while the LCV method identified several modifiable risk factors, further validation through experimental studies or randomized clinical trials is needed to confirm the causal nature of these associations. Fifth, the brain imaging and proteomic findings warrant further

investigation to fully elucidate the mechanisms by which these biomarkers and brain structures influence the shared genetic factor. Longitudinal studies integrating neuroimaging, proteomics, and genetic data could provide a more comprehensive understanding of the temporal and causal relationships underlying mvFibroPsych. Finally, although we initially termed the latent construct mvFibroPsych to reflect our motivation of exploring the shared genetic architecture between fibromyalgia and psychiatric traits, this factor should not be interpreted as specific to fibromyalgia. Rather, it represents a broader latent dimension of shared genetic liability encompassing fibromyalgia, depression, anxiety, and insomnia—closely resembling an internalizing or negative affect factor. Accordingly, the biological and functional annotations identified for this latent construct should be viewed as capturing mechanisms relevant to this transdiagnostic vulnerability domain, rather than to fibromyalgia per se.

## Conclusions

This study identifies a common genetic factor that underlies fibromyalgia, insomnia, depression, and anxiety, providing valuable insights into the associated biological pathways and risk factors. The identification of brain structural alterations and potential blood biomarkers, such as *VWC2*, presents promising avenues for future research and clinical applications. Additionally, multi-layer QTL analyses prioritize *CD40* as a potential target for addressing mvFibroPsych. These findings paves the way for more targeted therapeutic approaches and underscore the necessity for further research into the neurobiological mechanisms that contribute to the comorbidity between fibromyalgia and psychiatric disorders.

## Supporting information

**S1 Table. Detailed information for the included phenotypes.**
(XLSX)

**S2 Table. Genetic correlations among Fibromyalgia, Insomnia, Depression, and Anxiety.**
(XLSX)

**S3 Table. Sensitivity Analyses of Alternative GenomicSEM Model Configurations.**
(XLSX)

**S4 Table. Lead SNPs and Genomic Loci Associated with mvFibroPsych.**
(XLSX)

**S5 Table. Cross-Trait Z-scores and Heterogeneity Statistics for Genome-Wide Significant SNPs.**
(XLSX)

**S6 Table. SNPs in LD with any of independent significant loci with $r^2$ greater or equal to 0.6.**
(XLSX)

**S7 Table. Gene Prioritization of Protein-Coding Genes Associated with mvFibroPsych.**
(XLSX)

**S8 Table. Pathway Enrichment from MAGMA Gene-Set Analyses for mvFibroPsych.**
(XLSX)

**S9 Table. Potential Causal Relationships with mvFibroPsych Identified by LCV Analysis.**
(XLSX)

**S10 Table. Associations between brain structual MRI phenotyes and mv mvFibroPsychPsych.**
(XLSX)

**S11 Table. Associations between Brain diffussion MRI phenotypes and mvFibroPsych.**
(XLSX)

**S12 Table. Associations between mvFibroPsych and brain structural MRI phenotypes.**
(XLSX)

**S13 Table. Associations between mvFibroPsych and brain diffussion MRI phenotypes.**
(XLSX)

**S14 Table. Associations between Plasma protein levels and mvFibroPsych in the UKB-PPP datasets.**
(XLSX)

**S15 Table. Associations between Plasma protein levels and mvFibroPsych in the deCODE datasets.**
(XLSX)

**S16 Table. Associations between Plasma protein levels and mvFibroPsych in the FENLAND datasets.**
(XLSX)

**S17 Table. Associations between Plasma protein levels and mvFibroPsych in the INTERVAL datasets.**
(XLSX)

**S18 Table. Associations between brain gene expressions and mvFibroPsych.**
(XLSX)

**S19 Table. Association of Brain DNA Methylation with mvFibroPsych.**
(XLSX)

**S20 Table. Association of Brain Splicing QTLs with mvFibroPsych.**
(XLSX)

## Acknowledgments

The authors thank all investigators for sharing summary statistics of all GWASs included in this work.

## Author contributions

**Conceptualization:** Liling Lin, Yanni Fu.

**Formal analysis:** Yankai Li, Fengtao Ji.

**Investigation:** Fengtao Ji.

**Methodology:** Liling Lin.

**Resources:** Yanni Fu.

**Software:** Yankai Li.

**Supervision:** Liling Lin, Minghui Cao, Ganglan Fu, Yanni Fu.

**Validation:** Jianwei Lin, Diefei Liang, Minghui Cao.

**Visualization:** Yankai Li, Mengyi Zhu.

**Writing – original draft:** Liling Lin.

**Writing – review & editing:** Ganglan Fu.

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
