## [Decision Letter · Decision Letter 0]

25 Apr 2025

PGENETICS-D-25-00036

Uncovering the Psychiatric Underpinnings of Fibromyalgia: Novel Insights from Genomic Structural Equation Modeling

PLOS Genetics

Dear Dr. Lin,

Thank you for submitting your manuscript to PLOS Genetics. After careful consideration, we feel that it has merit but does not fully meet PLOS Genetics's publication criteria as it currently stands. Therefore, we invite you to submit a revised version of the manuscript that addresses the points raised during the review process.

Please submit your revised manuscript within 60 days Jun 24 2025 11:59PM. If you will need more time than this to complete your revisions, please reply to this message or contact the journal office at plosgenetics@plos.org. Please include the following items when submitting your revised manuscript:

We look forward to receiving your revised manuscript.

Kind regards,

Renato Polimanti, Ph.D.

Academic Editor

PLOS Genetics

Giorgio Sirugo

Section Editor

PLOS Genetics

Aimée Dudley

Editor-in-Chief

PLOS Genetics

Anne Goriely

Editor-in-Chief

PLOS Genetics

**Additional Editor Comments:**

While both reviewers appreciated the study described by the authors, they also highlighted several major concerns that should be fully addressed before considering the manuscript for publication in PLOS Genetics.

**Journal Requirements:**

At this stage, the following Authors/Authors require contributions: Liling Lin, Yankai Li, Fengtao Ji, Jianei Lin, Mengyi Zhu, Ganglan Fu, and Yanni Fu. Please ensure that the full contributions of each author are acknowledged in the "Add/Edit/Remove Authors" section of our submission form.

The list of CRediT author contributions may be found here: https://journals.plos.org/plosgenetics/s/authorship#loc-author-contributions

4) Please amend your detailed Financial Disclosure statement. This is published with the article. It must therefore be completed in full sentences and contain the exact wording you wish to be published. Please ensure that the funders and grant numbers match between the Financial Disclosure field and the Funding Information tab in your submission form. Note that the funders must be provided in the same order in both places as well.

**Reviewers' comments:**

Reviewer's Responses to Questions

**Comments to the Authors:**

Reviewer #1: Overview:

The authors aim to investigate genetic underpinnings related to chronic pain and psychiatric disorders by leveraging the genetic overlap among four GWAS summary statistics, including fibromyalgia, insomnia, depression, and anxiety disorder. They utilize the Genomic Structural Equation Modeling method to conduct a common factor analysis with these four traits, resulting in a combined genetic factor (mvFibroPsych). After verifying the fit indices of this model, they conduct thorough follow up analysis (gene, gene-set, tissue enrichment, latent causal variable method, QTL analysis, brain- and proteome-wide based mendelian randomization) to annotate genetic effects and interpret these as underlying the shared etiology among the four disorders.

The research is comprehensive insofar as it gathers available data from multiple cohorts and combines these in GWAS meta-analyses to increase power for their research design. However, I have fundamental concerns about this research design and how it pertains to the authors’ research questions.

Major comments:

-Across the four disorders, the number of GWAS cases as listed in supplementary table 1 amounts to 216,133 cases. Out of these, fibromyalgia cases amount to 2% (4,528) cases. Any latent factor comprised of these four traits is thus likely to mainly reflect genetic liability for the most powered traits (insomnia, depression, anxiety) and be less relevant for fibromyalgia. This is further exacerbated by the fact that only about half of the genetic variance of fibromyalgia is explained by the common factor model (i.e., the residual variance of fibromyalgia in the common factor model is 46%). Despite of this, the authors frequently interpret the results in the discussion as equally relevant to “chronic pain and psychiatric disorders”. I am hesitant of their interpretation as the result associated with the common factor will reflect the statistical power of the traits they include, of which fibromyalgia is the lowest powered one. Additional sanity checks are needed to verify that the author's interpretation is valid.

1.Investigate the individual contribution of each trait to the reported results for all follow-up analyses by conducting and presenting the same analysis on each of the four traits separately. If the effect of fibromyalgia is not nominally significant for e.g., a result in the QTL analysis then a more likely interpretation is that the effect is completely driven by higher power of depression/insomnia/anxiety and not relevant for fibromyalgia.

2.How does the presented results differ if you only run the common factor model with depression, insomnia and anxiety? If there is little difference between when fibromyalgia is or is not included in the model, then the respective effects are likely not relevant for chronic pain.

-Additional alternative genomic SEM models should be attempted, e.g., fix the factor loading of fibromyalgia to be completely explained by the latent factor. Another idea is to run pairwise models with fibromyalgia and depression / anxiety / insomnia separately to avoid findings being driven by two highly correlated psychiatric disorders.

-In UK Biobank there additional cases of fibromyalgia present. UKB was used in meta-analyses of insomnia/depression/anxiety, why not extend the fibromyalgia GWAS meta-analysis with UKB sample? For question Ever had Fibromyalgia syndrome (data field 120009) there are 3,010 additional cases. Alternatively, going by health records, there are 4,891 UKB participants diagnosed with Fibromyalgia (ICD10 code M79.7).

Minor comments:

-In Figure 2A and in-text: I miss standard errors and p-values of the genetic correlations.

-Figure 2C: novel factor hits; what are these 'novel' in relation to? Have none of these hits been associated with any of the included traits? What is the definition of indicator hits?

-Page 20, line 416-418: I find this claim circular. Is not the positive association between the mvFibroPsych and mood disorders expected as depression feature heavily in the latent model?

Reviewer #2: This study explores the genetic and neurobiological architecture of fibromyalgia, insomnia, depression, and anxiety, using GenomicSEM to identify a latent common factor (mvFibroPsych) and conducting GWAS, Mendelian Randomization (MR), and LCV analyses to assess causal relationships. The authors integrate data from UK Biobank, FinnGen, iPSYCH, and BioMe Biobank to strengthen their findings.

Their brain-wide MR analysis highlights a negative causal association between fractional anisotropy in the splenium of the corpus callosum and mvFibroPsych, suggesting white matter integrity in this region may have a protective effect against these comorbid conditions. This underscores the neurobiological component of shared genetic risk.

While the study is methodologically rigorous and addresses an important research question, a few aspects require further clarification to ensure robustness and interpretability.

Below, I outline key concerns and recommendations to improve clarity and validity.

1- The rationale for selecting the common pathway model over alternative pathway models is not clearly justified. Since various models could explain the genetic architecture of these phenotypes, the authors should provide a stronger empirical or literature-based justification for their choice.

I suggest two possible approaches to strengthen this aspect:

Empirical Model Comparison: The authors could compare multiple twin models (ACE, Cholesky decomposition, independent pathway, and common pathway) to determine which best explains the association between the phenotypes. This would provide a data-driven rationale for selecting the common pathway model.

Supporting Literature: If prior twin analyses support the use of the common pathway model for these traits, referencing relevant studies in the introduction and methods would help justify this choice.

Clarifying this would enhance methodological transparency and reinforce the robustness of the model selection.

2- The authors used UK Biobank, iPSYCH, and FinnGen for insomnia, depression, and anxiety GWAS summary statistics. However, larger meta-GWAS datasets exist for these traits, such as ISGC (insomnia), PGC-MDD (depression), and PGC (anxiety). Could the authors clarify why these were not used? If unavailable, discussing this limitation and its potential impact on the findings would improve transparency.

3- The authors conducted a meta-analysis combining UK Biobank and FinnGen data instead of using FinnGen as an independent replication cohort. This raises concerns about potential sample-specific biases and the ability to test the robustness of findings in an external dataset. Could the authors clarify why FinnGen was included in the meta-analysis rather than being used for replication? If independent replication was not feasible, discussing this limitation and its implications would improve transparency.

4- The genomic control factor (λGC = 1.207) and LDSC intercept (0.835, SE = 0.013) indicate a discrepancy—inflated test statistics despite a deflated intercept. While some inflation is expected in polygenic traits, this pattern may suggest heterogeneity across datasets rather than true polygenicity. Could the authors report the attenuation ratio (LDSC Intercept - 1 / λGC - 1) to assess whether inflation is driven by true polygenicity or residual confounding? If the ratio is high, it would support polygenic architecture as the primary driver of inflation.

5-The authors applied both Mendelian Randomization and Latent Causal Variable analysis but did not clarify their rationale for selecting one method over the other for specific exposures. Since MR estimates direct causal effects while LCV assesses genetic causality via genetic correlations, could the authors specify their criteria for choosing between these approaches? Was the availability of strong instrumental variables a determining factor? A brief justification in the methods section would improve clarity and transparency.

6-The reported CFI = 0.999 suggests an almost perfect model fit, which may indicate overfitting or an imbalanced contribution of traits to the common factor. Given that depression has the largest sample size and highest factor loading, could the authors clarify whether this might be driving the model’s fit? Additionally, have they assessed whether sample size disparities across traits (e.g., depression vs. fibromyalgia) could be influencing the covariance structure? A sensitivity analysis adjusting for sample size effects would help ensure that the model is not overly weighted by depression

7- The authors report significant SNPs from their common factor GWAS but do not mention whether they assessed heterogeneity in SNP effects across the component traits. Given that some SNPs may exhibit trait-specific effects rather than acting through the shared factor, did the authors perform any heterogeneity tests, such as Cochran’s Q statistic or variance partitioning across phenotypes? If not, a brief discussion of potential heterogeneity would improve transparency.

8-Many of the genome-wide significant SNPs have extremely small p-values, which may be influenced by differences in GWAS sample sizes across traits. Since depression GWAS has the largest sample size, could the authors clarify whether depression-associated SNPs disproportionately contribute to the common factor model loadings?

Reviewer #3: This article explores the shared genetics and molecular biology between fibromyalgia, anxiety, depression, and insomnia. Fibromyalgia is an idiopathic condition characterized by widespread musculoskeletal pain, and is often comorbid with psychiatric conditions. This begs the question of whether there are shared genetic risks and biological pathways between these conditions. The authors use genomic structural equation modeling (gSEM) and genome-wide association studies (GWAS) to identify associated genetic variants that are shared between these phenotypes, and employ a suite of functional genomics tools to investigate the biology underlying these variants; including FUMA, Latent Causal Variable (LCV) Analysis, brain-wide and proteome-wide Mendelian Randomization (MR) and QTL analysis.

I have little domain knowledge of fibromyalgia but have expertise in psychiatric genetics, gSEM, GWAS, and all the downstream analyses except LCV and MR.

Overall, the article is very well-written and has excellent flow. The authors also do a great job of explaining the purpose of each analysis and how to interpret the results from each one. My major comments center around including more information about the GWAS analyses.

ABSTRACT AND INTRODUCTION

Major issues:

Minor issues:

-Clarify that the meta-analyses were within phenotypes, not across phenotypes (see comment in minor issue for methods)

-Provide more detail about the GWAS including ancestry, and effective sample size.

-Were the 64 novel loci novel for fibromyalgia, the psych conditions, or both?

-In the introduction, please include a paragraph that summarizes previous genetic studies of fibromyalgia, including any recent GWAS

METHODS

Major issues:

-There is no information on how the GWAS of mvFibroPsych was run. Please add a paragraph for this topic and include information such as the effective sample size, what software was used to run the analysis, what covariates were used, whether a logistic or linear regression was used, etc.

Minor issues:

-It is my understanding that you did a meta-analysis of two studies for each phenotype, then fed these four sets of sumstats into LDSC and gSEM. This is not what you describe; rather, you describe doing a meta-analysis ACROSS all phenotypes to generate a single set of sumstats, which was then fed into LDSC and gSEM, which is not possible. Please clarify this point.

-I know you include the sample sizes for the individual GWAS in the supplement, but please include the effective N for each meta analysis results, and the gSEM GWAS result, in the Methods. The reader will want to know.

-In the GWAS Meta-analysis section, please include a description of how the phenotypes were defined in the GWAS. Were they binary (case-control) phenotypes? How were cases defined, given that you mention earlier that fibromyalgia is heterogenous?

-For the genome-wide association study annotation section, please mention what % of variants were assigned by FUMA to their effector gene based on positional mapping vs which had functional evidence (QTL, chromatin data) supporting their gene assignment

RESULTS

Major issues:

-At the beginning of the results section, please include a paragraph that summarizes the results of the four GWAS meta-analyses you did. What were the resulting sample sizes for each of the four phenotypes? Did performing the meta-analyses increase the number of significantly associated loci? You mention that you did the meta-analyses to improve power, but don’t say anything about whether they actually did improve power.

Minor issues:

-Typically when a GWAS is run on a gSEM model, there is one paragraph about the gSEM model results and one paragraph about the GWAS results. I would recommend following this format by ending the first paragraph with the sentence ‘the standard factor loadings...” and beginning a new paragraph with the next sentence ‘The common factor model built in the gSEM model’. I would move the first sentence of the next paragraph ‘A total of 122 lead SNPs’ into this new paragraph as well.

-In the latent common factor GWAS estimation paragraph, please comment on whether the observed genetic correlations are reasonable/expected based on previous studies

-Please clarify what you mean by novel when referring to the 64 novel loci. Were they novel associations for fibromyalgia, the psych disorders, or all?

-The figures are blurry, please provide clearer images

-In Figure 2c, I would suggest using the terms ‘lead SNPs’ and ‘novel lead SNPs’ instead of ‘indicator hits’

-The description of the LVC results in the result section is a bit confusing; it seems like both descriptions are describing mvFibroPsych causally influencing something:

--‘Among these, 58 phenotypes were found to be causally influenced by mvFibroPsych, whereas mvFibroPsych was identified as a causal factor for 13 phenotypes’

-Please describe what ansiotropic diffusion is

-Please make sure to italicize gene names

DISCUSSION

Major issues:

-Please discuss the gSEM model and whether the loadings make sense. Why is that insomnia loaded significantly less onto mvFibroPsych despite having a similar genetic correlation to fibromylagia, anxiety, and depression?

-Please discuss the GWAS results in the context of previous fibromyalgia genetic studies – did you replicate any previous signals? What loci were new? It would be interesting to include a pathway analysis on just the new loci

Minor issues:

**Have all data underlying the figures and results presented in the manuscript been provided?**

Reviewer #1: **No:** Standard errors and p-values of the genetic correlations presented in figure 2A are not provided.

Reviewer #2: Yes

Reviewer #3: Yes

PLOS authors have the option to publish the peer review history of their article (what does this mean? ). If published, this will include your full peer review and any attached files.

**Do you want your identity to be public for this peer review?** For information about this choice, including consent withdrawal, please see our Privacy Policy .

Reviewer #1: **Yes:** Cato Romero

Reviewer #2: No

Reviewer #3: No

**Figure resubmission:**
---

## [Decision Letter · Decision Letter 1]

21 Aug 2025

PGENETICS-D-25-00036R1

Uncovering the Psychiatric Underpinnings of Fibromyalgia: Novel Insights from Genomic Structural Equation Modeling

PLOS Genetics

Dear Dr. Lin,

Thank you for submitting your manuscript to PLOS Genetics. After careful consideration, we feel that it has merit but does not fully meet PLOS Genetics's publication criteria as it currently stands. Therefore, we invite you to submit a revised version of the manuscript that addresses the points raised during the review process.

Please submit your revised manuscript within 60 days Oct 20 2025 11:59PM. If you will need more time than this to complete your revisions, please reply to this message or contact the journal office at plosgenetics@plos.org. Please include the following items when submitting your revised manuscript:

We look forward to receiving your revised manuscript.

Kind regards,

Renato Polimanti, Ph.D.

Academic Editor

PLOS Genetics

Giorgio Sirugo

Section Editor

PLOS Genetics

Aimée Dudley

Editor-in-Chief

PLOS Genetics

Anne Goriely

Editor-in-Chief

PLOS Genetics

**Reviewers' comments:**

Reviewer's Responses to Questions

**Comments to the Authors:**

Reviewer #1: I thank the authors for providing additional information regarding their submission as requested.

In regard to their response to my previous comments, I have remaining disagreements, in addition to new concerns upon examining the provided supplementary information, particularly supplementary table S4 and S5.

**Major comments:**

1)

The Manhattan plot in figure 2C shows the smallest SNP p-value of the latent mvFibroPsych factor to be ~1e-30. However, in supplementary table S4 ‘Lead SNPs and Genomic Loci Associated with mvFibroPsych’, multiple SNPs have a p-value lower than 1e-200. There is an inconsistency in what is plotted in the main figure of figure 2C and what is reported in the supplementary tables S4 and S5.

Furthermore, supplementary table S5 contains theoretically impossible results, such as for SNP rs979343, which shows a mvFibroPsych Z score of -31.050, while the input Z score of anxiety, depression, insomnia, and fibromyalgia range from -0.3 to 0.35.

Similar incoherent results are reported for rs7067621, rs6783689, rs667255, rs6459268, rs6424719, rs61023573, rs59608961, rs4940514, rs17797882, and rs150011668.

In my estimation, these SNP effects reported in supplementary table S4 and S5 are unreliable and must be investigated further. Alternatively, the correct estimates must be reported again in supplementary table S4 and S5, and be resubmitted.

2)

In response to my concern that fibromyalgia cases amount to 2% of the total case number across the 4 included traits in the study, the authors argue that GenomicSEM is effective in boosting the genetic signal of less powered traits by utilizing the genetic covariance with other, well-powered traits. They refer to publications by Rosoff et al. 2023, Grotzinger et al. 2019, and Chen et al. 2023, to underline how this has been applied in previous studies.

To start, I do not see a mention of GenomicSEM in the Chen et al. 2023 publication “Genomic atlas of the plasma metabolome prioritizes metabolites implicated in human diseases”, so it is unclear to me what I am supposed to take away from this article.

I agree with the authors that conducting GenomicSEM on a set of traits with varying sample sizes and statistical power is possible, however, this study design has consequences for the final interpretation of the results. It is in this interpretation that I find the way the discussion section in the paper is currently lacking.

To illustrate my point, the Rosoff et al. paper combines GWASs of 5 aging-related traits (frailty, healthspan, parental lifespan, longevity, and phenotypic age acceleration) into one general aging related latent factor. All functional annotations associated with this latent factor is interpreted as generally relevant for aging-related processes.

In the current paper, GWAS of fibromyalgia, depression, anxiety and insomnia are combined into the latent factor mvFibroPsych and significant functional annotations are consistently contextualized as fibromyalgia AND psychiatric disorders. See examples: “[…]comprehensive understanding of the shared genetic underpinnings of fibromyalgia and its associated psychiatric disorders”(p.7 line 136), or “[…]neurobiological mechanisms that contribute to the comorbidity between fibromyalgia and psychiatric disorders” (p30, line 642).

Hence when the authors are interpreting the results associated with the latent factor mvFibroPsych, they refer to the observed factors that make up the latent factor instead of interpreting the latent factor itself. This seems to me to be a fallacy in interpreting the current study design, which leads to inaccurate reporting given the underlying evidence. The latent factor mvFibroPsych might as well be termed mvInternalising or mvNegativeAffect and the connection to chronic pain would already seem less obvious.

In the very least, I require that this features more prominently in their discussion of study limitations and that reporting of results are deemed putative unless confirmed in additional SNP-to-latent factor analyses.

3)

While I appreciate the authors efforts in conducting alternative GenomicSEM models showing Fibromyalgia with each of the three psychiatric disorders separately, sufficient model fit showing shared genetic covariance is not the main point. The main point is the resulting SNP associations of these separate latent models and their subsequent association with functional annotations. I still am not convinced that the majority of the reported results are at all relevant to fibromyalgia. This is underlined by the fact that among the 122 reported lead SNPs in the mvFibroPsych latent factor, 81 SNPs do not show nominal significance (Z score range from 1.96 to -1.96) in the individual Fibromyalgia GWAS, as seen in Supplementary table S5. In addition, for many of the 122 genome-wide significant lead SNPs there are discordance in the effect direction among the individual traits, which makes interpretation less straight forward as some downstream associations seem protective for some traits while risk increasing for others.

**Minor comments:**

1)

On page 20, line 410-413, the authors write “[…]gene-set analyses revealed that mvFibroPsych has an enrichment in the pathways involved in synaptic function, AMPA glutamate signaling, benzodiazepine receptor, and potassium ion transporters activities (Table S8).” Table S8 shows that these gene-sets, except GOCC_SYNAPTIC_MEMBRANE, are not significant after bonferroni correction and should not be reported.

**Have all data underlying the figures and results presented in the manuscript been provided?**

Reviewer #1: Yes

PLOS authors have the option to publish the peer review history of their article (what does this mean? ). If published, this will include your full peer review and any attached files.

**Do you want your identity to be public for this peer review?** For information about this choice, including consent withdrawal, please see our Privacy Policy .

Reviewer #1: No

**Figure resubmission:**
---

## [Decision Letter · Decision Letter 2]

16 Nov 2025

PGENETICS-D-25-00036R2

Uncovering the Psychiatric Underpinnings of Fibromyalgia: Novel Insights from Genomic Structural Equation Modeling

PLOS Genetics

Dear Dr. Lin,

Thank you for submitting your manuscript to PLOS Genetics. After careful consideration, we feel that it has merit but does not fully meet PLOS Genetics's publication criteria as it currently stands. Therefore, we invite you to submit a revised version of the manuscript that addresses the points raised during the review process.

We look forward to receiving your revised manuscript.

Kind regards,

Renato Polimanti, Ph.D.

Academic Editor

PLOS Genetics

Giorgio Sirugo

Section Editor

PLOS Genetics

Aimée Dudley

Editor-in-Chief

PLOS Genetics

Anne Goriely

Editor-in-Chief

PLOS Genetics

**Additional Editor Comments:**

Reviewer #1 raised some remaining concerns regarding some of the findings. I agree with their evaluation, especially the one regarding the presence of suspicious genetic associations.

Accordingly, I strongly recommend that the authors investigate these further and eliminate them if they are spurious findings.

**Journal Requirements:**

1) We have noticed that you have uploaded Supporting Information files, but you have not included a list of legends. Please add a full list of legends for your Supporting Information files after the references list.

2) As required by our policy on Data Availability, please ensure your manuscript or supplementary information includes the following:

**Reviewers' comments:**

Reviewer's Responses to Questions

**Comments to the Authors:**

Reviewer #1: I much appreciate the changes made to this version of the paper. Thank you to the authors for thoroughly repeating their work to increase the robustness and reliability of the work they present. I particularly think that the discussion has improved and presents sufficient nuance and statistical caution with regards to the interpretation of the results. I only have two remaining minor comments, after which I would regard this paper ready to be suitable for publication.

1) The authors have thoroughly rephrased interpretation of results from their study design, replacing formulations like "significant for fibromyalgia and its associated psychiatric disorders” with “significant for the mvFibroPsych latent factor, representing shared genetic liability across fibromyalgia and psychiatric traits.”. This is good, but I would argue that the title of the study ("Uncovering the Psychiatric Underpinnings of Fibromyalgia:[...]") still assumes that presented findings are all associated with fibromyaligia and the other psychiatric disorders. I would suggest to change the title to avoid this assumption.

2) Looking at the Manhattan plot in figure 2C: the top SNP (rs4865477, p-value = 2.31e-24) is highly suspicious. Suppl Table S6 shows other SNPs in >0.6 LD with this SNP having p values of ~0.045. Additionally in Supple Table S5, the input Z-scores of the 4 traits for rs4865477 are all between -1.96 and 1.96, ie. not nominally significant, yet somehow combining these input z-scores produces the most significant association with the latent factor.

Though it is a bit difficult see from figure 2C, several of the other novel hits (on chr 23, chr 9 and chr 8 appear to be significant without any surrounding SNPs showing signs of significance as well. These singular significant SNPs should be carefully investigated or removed outright from results and downstream analysis, as they are biologically implausible.

**Have all data underlying the figures and results presented in the manuscript been provided?**

Reviewer #1: Yes

PLOS authors have the option to publish the peer review history of their article (what does this mean? ). If published, this will include your full peer review and any attached files.

**Do you want your identity to be public for this peer review?** For information about this choice, including consent withdrawal, please see our Privacy Policy .

Reviewer #1: No

**Figure resubmission:**
---

## [Editor Report · Decision Letter 3]

14 Jan 2026

Dear Dr Lin,

We are pleased to inform you that your manuscript entitled "Shared Latent Genetic Liability Across Fibromyalgia and Psychiatric Traits : Novel Insights from Genomic Structural Equation Modeling" has been editorially accepted for publication in PLOS Genetics. Congratulations!

Yours sincerely,

Renato Polimanti, Ph.D.

Academic Editor

PLOS Genetics

Giorgio Sirugo

Section Editor

PLOS Genetics

Aimée Dudley

Editor-in-Chief

PLOS Genetics

Anne Goriely

Editor-in-Chief

PLOS Genetics

BlueSky: @plos.bsky.social

Comments from the reviewers (if applicable):

**Data Deposition**

http://datadryad.org/submit?journalID=pgenetics&manu=PGENETICS-D-25-00036R3

**Press Queries**

---

## [Editor Report · Acceptance letter]

PGENETICS-D-25-00036R3

Shared Latent Genetic Liability Across Fibromyalgia and Psychiatric Traits : Novel Insights from Genomic Structural Equation Modeling

Dear Dr Lin,

We are pleased to inform you that your manuscript entitled "

Shared Latent Genetic Liability Across Fibromyalgia and Psychiatric Traits : Novel Insights from Genomic Structural Equation Modeling " has been formally accepted for publication in PLOS Genetics! Your manuscript is now with our production department and you will be notified of the publication date in due course.

With kind regards,

Judit Kozma

PLOS Genetics

On behalf of:
